# A Review on Thermal Conversion of Plant Oil (Edible and Inedible) into Green Fuel Using Carbon-Based Nanocatalyst

**G. Abdulkareem-Alsultan [1,*], N. Asikin-Mijan [1,2], H. V. Lee [2], Umer Rashid [3], Aminul Islam [4] and Y. H. Taufiq-Yap [1,*]**

1   Catalysis Science and Technology Research Centre (PutraCat); Faculty of Science, Universiti Putra Malaysia, Serdang 43400 UPM, Selangor, Malaysia; ckin_mijan@yahoo.com
2   Nanotechnology & Catalysis Research Centre (NanoCat), Institute of Postgraduate Studies, University Malaya, Kuala Lumpur 50603, Malaysia; leehweivoon@um.edu.my
3   Institute of Advanced Technology, Universiti Putra Malaysia, Serdang 43400 UPM, Selangor, Malaysia; umer.rashid@yahoo.com
4   Chemical Engineering Department, Jessore University of Science and Technology, Jessore 7408, Bangladesh; aminul03211@yahoo.com
*   Correspondence: kreem.alsultan@yahoo.com (G.A.-A.); taufiq@upm.edu.my (Y.H.T.-Y.);
    Tel.: +60182534058 (G.A.-A.); +60126293195 (Y.H.T.-Y.)

**Abstract:** Renewable diesels (e.g., biodiesel and green diesel) have emerged as a sustainable alternative to petrodiesel as a means of meeting the growing demand for fuel without damaging the environment. Although renewable diesels are composed of different chemical compositions to petrodiesel, they provide similar fuel characteristics as petrodiesel. The present articles focused on various type of green diesel, where the properties and its performance are discussed in detail. Green diesels offer multiple benefits over petrodiesel in terms of biodegradability, environmental protection and low toxicity. Additional, this paper described various types of process for green diesels production such as deoxygenation, hydrodeoxygenation, and pyrolysis. Among the synthesis process, the most effective and economical route to produce green diesel is through deoxygenation (DO). This study also emphasizes the use of a carbon-based catalyst for the DO reaction. The carbon-based catalyst renders several advantageous in term of highly resistance toward coke formation, greater catalyst stability, and product selectively, where the DO process occur via carbon–oxygen cleavage of fatty acid chain yielding diesel-like hydrocarbons. Due to this reason, various methods for synthesizing effective carbon-based catalysts for the DO reaction are further reviewed. Coke affinity over carbon-base catalyst during DO process is further discussed in the present study. Besides, DO operating condition toward optimum yield of hydrocarbons and recent progress in DO of realistic oil for production of diesel-like hydrocarbons are also discussed herein.

**Keywords:** green diesel; renewable diesel; deoxygenation; sustainable fuel; carbon-based catalyst

---

## 1. Introduction

The growing demand for energy and fuels in the developed country is well established, with predictive data showing a likely rise in consumption of diesel up to 10 million barrels per day from 2012 to 2035 [1]. This is in agreement with the International Energy Agency estimation, which predicts that the global energy demand is anticipated to rise by 53% by 2030. The major source of the energy in the world is currently fossil fuel (87%), in which crude oil accounts for 34%, coal for 31%, and natural gas for 24% [2]. Since the major part of energy accounts by the crude oil, thus it also raises

concerns over the exhaustion of fossil fuels as well as the environmental degradation by its continues usage. It is speculated that the fossil fuel reserves are depleting at rate of 4 billion tons a year. If this rate continues, oil deposits will be exhausted by 2052 [3]. Moreover, continued usage of fossil fuels also generally results in the anthropogenic emission of $CO_2$ into the atmosphere accelerating global climate change [4]. Therefore, various researches were performed to search an alternative substitute for conventional crude oil. Many renewable energy sources have drawn the attention of researchers. Example of alternative renewable energy sources is biomass.

The current renewable biomass-based alternative fuels available nowadays are biodiesel and green diesel. Although biodiesel have been established and continuously use by the society, but biodiesel composed of different chemical compositions than petrodiesel fuel. Biodiesel are fatty acid methyl ester (FAME), which compose of high viscosity as compared with petrodiesel. Thus, it poses a technological challenge that impractical for direct usage for diesel engine. Moreover, the existence of oxygenates and unsaturated C–C bonds in biodiesel could result in high freezing point, limited compatibility with gas engines, low chemical stability, and low caloric value in the biodiesel [5]. Due to this reason, the biodiesel will be blended with petrodiesel fuel at sufficient amounts based on the application [6]. This is in contrast to green diesel, which is composed of fuel-like hydrocarbons that are entirely fungible with petroleum derived fuels. The green diesel can be directly use in the engines without any modification to the engine or the fuel itself.

Several alternative techniques recently developed to produce green diesel using biomass-derived feedstock (fatty acid and triglycerides) including deoxygenation (DO), hydrodeoxygenation (HDO), and pyrolysis. All these techniques involve catalytic cracking technology whereby the high-molecular-mass compound is fragmented into a molecule with lower molecular mass [7]. The DO process involves the removal of oxygen in the form of a carboxyl group from a fatty acid or triglycerides under a hydrogen-free atmosphere. Meanwhile, HDO involves removal of oxygen via introduction of hydrogen gas under elevated pressure condition. Pyrolysis involves thermal method, converting biomass into gases, solid char, condensable pyrolytic vapor which higher value fuels (bio-oil) [8]. For instance, bio-oil contains high oxygen content (35–40%) and hundreds of organic compounds that belong to acids, alcohol, ketones, aldehyde, phenols, ester, sugar, etc. Thus, it would affect the homogeneity, polarity, heating value (HV), viscosity, and acidity of oil. Bio-oil also contains 15–30 wt % water, although up to 60 % is reported [9,10]. Therefore, high water content will reduce the immiscibility of the bio-oil with crude oil. Due to economical point of view and miscibility of liquid fuel produced, DO offer great advantages.

Numerous study found metal-supported catalyst (ex: CaO-La$_2$O$_3$/AC$_{nano}$, CoFeN$_x$/C, Pd/C, Pt/C, Ni/MWCNT, NiCo/MWCNT, Ag$_2$O$_3$-La$_2$O$_3$/AC$_{nano}$) facilitated DO with product selectively toward diesel range hydrocarbons [11–15]. However, metal-supported catalyst typically results in diminished activity due to the coke formation. Therefore, the use of carbon support in DO reaction has been widely studied and majority of the DO reaction over carbon supported catalyst showed in very encouraging results [16,17]. Other than catalyst, it also reported that DO operating parameter also play important role in optimizing the hydrocarbon yield and diesel-like hydrocarbon selectivity in DO reaction. The most important operating parameters include feed type, reaction atmosphere, and reaction temperature. This review summarizes the multiple reactions for the production of green diesel and the use of carbon supported catalysts is highlighted. The role of feedstocks, reaction atmosphere, and reaction temperatures in optimizing the DO were discussed. The paper ends with an overview towards state-of-the-art technology in renewable fuels.

## 2. Green Diesel vs. Biodiesel

In comparison to petrodiesel, both biodiesel and renewable diesel have benefits with respect to carbon renewability. However, when compared in terms of fuel properties, environmental effect, and energy stability, each has certain relative strengths over the other. Biodiesel fuel mainly contains mono-alkyl esters of long-chain fatty acids, obtained from vegetable or animal sources

by a transesterification reaction of oil or fat, with glycerol as a coproduct. In comparison, green diesel originates from biological sources only. The term "green diesel" is somewhat misleading, however, as this implies that green diesel is more environmentally protective than other kinds of diesel. In reality, the term originates from the production of the fuel and its positive effects on greenhouse gas build up, and it should not be taken to imply improved performance in terms of issues such as biodegradability.

A comparison of the chemical quality of the two fuel types was conducted by Kalnes and Marker, as shown in Table 1 [18].

**Table 1.** Comparison of fuel quality: biodiesel vs. green diesel [18].

| Fuel Properties | Biodiesel | Green Diesel |
|---|---|---|
| Oxygen, % | 11 | 0 |
| Specific gravity | 0.88 | 0.78 |
| Sulfur, ppm | <1 | <1 |
| Heating value, MJ/kg | 38 | 44 |
| Cloud point, °C | −5 to +15 | −20 to +20 |
| Cetane | 50–65 | 70–90 |
| Stability | Marginal | Good |

Green diesel has more heating value and stability than biodiesel, and the two fuels also display important differences with respect to emissions when used as fuel. Research has repeatedly highlighted the higher nitrogen oxide ($NO_x$) emissions of engines using biodiesel as a fuel as compared to green diesel [1]. However, with regard to the emission of other environment pollutants such as particulate matter (PM), hydrocarbons (HC) and carbon monoxide (CO), and the use of biodiesel in an engine results in significant reductions [1]. Nevertheless, with the use of novel emissions control technologies such as selective catalytic reduction (SCR) and the introduction of more advanced new engines, it is expected that the exhaust emissions of $NO_x$ and PM will be further reduced to meet environmentally acceptable standards irrespective of the fuel used [19,20]. Hence, the differences between these fuels with regard to the exhaust emission of pollutants will be less significant in the future.

It is also possible to compare the differences in exhaust emissions of pollutants with respect to different individual components of biodiesel and green diesel, however. In a study comparing the exhaust emissions from biodiesel and its components and petrodiesel, as well as neat alkanes [21], the emissions of a relatively new engine, a 14 L, six-cylinder, turbocharged, intercooled 2003 model with features such as exhaust gas recycling, were examined. That study found that the use of biodiesel and its components lowered exhaust emissions of PM by 73 to 83% in comparison to petrodiesel fuel, with a difference of 45 to 50% emerging when compared with neat alkanes [21]. The exhaust emission of $NO_x$ was highest for methyl acrylate, with increases of up to 12.5%, and lowest for hexadecane, where it showed a decrease of up to 15.7% in comparison to petrodiesel [21]. In the case of HC, a reduction in exhaust emissions was seen when alkanes were used instead of biodiesel and its components. However, biodiesel and its components had lower exhaust emissions of CO in comparison to the alkanes. Taken as a whole, the research shows that biodiesel is preferable to green diesel in terms of PM and CO exhaust emissions, but that green diesel offers better reduction of $NO_x$ exhaust emissions and a slight reduction in HC exhaust emissions. Other studies have also considered to the effects of compound structures such as double bonds and chain length, and similar conclusions emerged [18,22,23].

A number of other studies have also shown benefits for using green diesel over other fuels [18,24–27]. These studies conducted emission tests to compare green diesel with other fuels, using a petrodiesel fuel that satisfied European petrodiesel standard EN 590 [24]. These studies found that the use of green diesel reduced $NO_x$ exhaust emissions by 7 to 14%, PM by 28 to 46%, CO by 5 to 78%, and HC from 0 to 48% in comparison to standardized petrodiesel fuel [25,26]. Some studies also compared the two types of non-petrodiesel with respect to their capacity to cause mutation of exhaust emissions [27–30]. While earlier studies reported biodiesel as having more benefits than green diesel in this regard [28,29], more recent studies have reported important advantages for green

diesel [27], while one study found that petrodiesel satisfying EN 590 requirements and biodiesel produced similar effects [30]. However, the use of biodiesel does allow a reduction in the number of particles in comparison to either petrodiesel or green diesel [27].

Studies comparing the biodegradability of different types of diesel have shown that biodiesel generally decomposes more speedily under suitable conditions with appropriate microbes, notwithstanding tests of blends of biodiesel with petrodiesel to boost biodegradation [31–33]. However, to the author's knowledge, no study has yet compared the biodegradability of biodiesel and green diesel. Knothe [1] is of the opinion that green diesel might have a decomposition ability equal to that of petrodiesel because they share similar compositions; however, green diesel's n-alkanes might provide better performance in terms of biodegradability in comparison to petrodiesel or green diesel with other compositions.

Another important environmental concern that might affect the choice between biodiesel and green diesel is amount of deforestation caused by each. Some researchers have raised concerns over the deforestation of palm trees in tropical forests for the production of biodiesel [31]; however, palm oil is also a major source for green diesel and hence the effect is similar [1]. A possible way to address this problem is the increased use of waste oil for the production of green diesels [34].

Finally, cost considerations also influence the choice between green and bio diesels. Knothe [1] compared the cost of green diesel with biodiesel and reported them as having similar costs of production, as the production process in both cases can make use of the same feedstocks, which is the most costly element in the process. For biodiesel, the cost of lipid feedstock is the main source of higher costs in comparison to petroleum. Green diesel has similarly higher cost because triglycerides are the main source used. Determining the exact difference in cost of production between the two non-petrodiesels is deemed difficult by Knothe [1] due to lack of hard data about the cost of production of green diesel. Additionally, the use of waste material for the production of green diesel can make it more cost-effective [34].

Both green diesel and biodiesel have their own advantages and hence, the choice between the two must be based on the specific use case as well as the requirements of the area where it is to be used. Green diesel offers multiple benefits over petrodiesel in terms of environmental protection and some advantages over biodiesel with regard to the exhaust emission of certain pollutants.

## 3. Alternative Pathways for Production of Green Diesel

A number of strategies have been proposed for the production of green diesel. The three main alternative strategies—deoxygenation, hydroxygenation, and pyrolysis—are thus discussed in more detail below, including an outline of the reaction pathway of each, as provided in the research.

### 3.1. Deoxygenation (DO)

The main rationale behind the use of deoxygenation (DO) is to manage the oxygenated compounds in the feedstock, which are responsible for lowering chemical stability and decreasing the energy content of the final product. DO produces a liquid fuel with much higher stability and better combustion properties through a multiple reaction process that includes steps such as dehydration, decarboxylation, and decarbonylation. The process of dehydration focuses on the –OH bonds, removing oxygen in the form of water. The decarboxylation process involves the removal of oxygen from carboxylates, making it particularly suitable for producing diesel with lower acidity. Finally, decarbonylation removes carbonyl groups from the product to improve its stability and reduce its heating value. The reaction pathway for the decarboxylation and decarbonylation of tristearin and steric acid is shown in Figure 1.

**Figure 1.** Reaction pathway of deoxygenation (DO) of (**a**) tristearin and (**b**) stearic acid [35].

Natural oils and fats are composite blends of triglycerides that contain three fatty acids and a glycerol half chain. Structurally, the saturated and unsaturated fatty acids in these natural sources usually have high carbon numbers, indicating high heating values. These high heating values are of major concern in regard to the feasibility of these natural oils as substitutes for petroleum, and hence it is critical to remove oxygenates and water from these oils to make them into usable diesel products. They are thus most frequently used as feeds for producing green diesel by means of selective deoxygenation [36,37]. Selective DO can be done through a process of decarboxylation and decarbonylation [38]. However, another recently proposed method, in which saturated feeds are deoxygenated in liquid form in the presence of heterogeneous catalysts such as Pd/C [39,40] under conditions of high temperature and pressure, appears to offer effective production. For unsaturated feeds, thermolysis in the presence of metal oxide catalysts has also been suggested [41].

### 3.2. HDO (Hydrodeoxygenation)

Hydrodeoxygenation is a key process for the production of green diesel through the removal of oxygenated compounds from feedstock. It requires a feedstock containing double bonds and oxygenated compounds, which is then converted into hydrocarbons through the saturation of the double bonds and the removal of oxygen. The process requires high temperature and pressure conditions as well as the presence of hydrogen as a coreagent [42,43]. The advantages of using this process include the production of a diesel with lower viscosity and higher lubricity; the absence of oxygen in the diesel also reduces its reactivity and makes it more stable.

The reaction pathway of the process using (a) tristearin and (b) stearic acid as a feed is shown in Figure 2. The process involves the conversion of the triglycerides present in natural feedstocks into *n*-alkanes, which generates water and propane gas as by-products. In terms of removal of oxygen from the triglycerides, hydrogen in the atmosphere breaks the C=O bond, C–O bond, and C–C bonds. The role of catalysts is to assist in the breaking of these bonds and these are most likely to influence the route and the degree of C–C and C–O bond cleavage [44,45].

A variety of catalysts can be used for this process including $NiMo/\Upsilon\text{-}Al_2O_3$ or $CoMo/\Upsilon\text{-}Al_2O_3$ [42–46] CoMo/C [47–49], CoMo/Si [44] $Pd/SiO_2$, Pd/C [41,45,50], and Pt/C [51], depending upon the hydroprocessing reaction required. While CoMo- and NiMo-based catalysts are most popular and have been in use for decades, recent studies have reported that their reactivity, when compared to newly introduced catalysts, is relatively low [46,52]. The traditional process involves vegetable oil as feedstock, which is hydroprocessed with NiMo- and CoMo-based catalysts

in a temperature range of 350 to 450 °C with a liquid hour space velocity of 0.5 to 5.0 per hour [53]. Modifications have thus been proposed in recent years with the introduction of new catalysts [45,50–54].

(a) Tristearin

(b) Stearic acid

**Figure 2.** Reaction pathways of hydrodeoxygenation (HDO) in (**a**) tristearin and (**b**) stearic acid [35].

Other common hydroprocessing reactions involved in the production of green diesel include hydronitrogenation (HDN), hydrodesulfurization (HDS), and hyrodematalisation (HDM). These reactions differ with respect to the element they target for removal from the feedstock, with HDN removing nitrogen, HDS removing sulfur, and HDM removing metal compounds [53,55]. The same catalysts can be used for HDO, HDS, or HDN reactions, however, due to the similarity of these processes. However, particularly in relation to green diesel, HDO of triglycerides is generally preferred.

*3.3. Pyrolysis*

Pyrolysis is an extreme version of thermal cracking followed by a consequent rearrangement of produced fragments. The green oil or bio-oil produced through the pyrolysis of biomass has the appropriate properties to be used as an alternative fuel source and it can also be used to synthesize other chemicals and products. Pyrolysis has been reported to be much simpler and less costly than other methods for the production of green diesel [56], and the hydrocarbon products produced through pyrolysis usually have shorter chains than products produced through HDO or DO [1]. The process requires temperatures as high as 300 to 400 °C under normal pressure, but it does not need hydrogen as a coreagent [1].

Several different kinds of pyrolysis reactions have been introduced in the literature that influence the yield and reaction conditions. Slow pyrolysis conducted under lower temperature conditions takes a longer time to yield oil [57], while high temperature pyrolysis yields oil in much less time [58]. A comparison of the reaction pathways of three different types of pyrolysis is provided in Figure 3 [59]. Onay and Kapoor [56] compared fixed-bed and fast pyrolysis on rapeseed feedstock, reporting higher yields of oil from the fast pyrolysis technique. Many other studies have corroborated these results, and it has been suggested that slow pyrolysis is most suited for the production of charcoal [60–62]. Other categories of pyrolysis include catalytic fast, intermediate, and vacuum [62]. Yields vary with respect to the choice of feedstock as well as the kind of reactor used, and there is a variety of reactors to choose from [62].

Research on pyrolysis of biomass sources has mainly neglected the triglyceride materials found to produce a green oil with different properties to the oil produced through other sources [34]. However, in recent years, a few studies have been conducted to examine the reaction pathways and oil production efficiency of pyrolysis using triglyceride materials [56–58]. The results of these studies indicate that, in

comparison to other methods for the production of green diesel through triglycerides, pyrolysis has lower costs and can be used with a wider variety of feedstock, including waste materials [34].

**Figure 3.** Reaction pathway of cellulose pyrolysis [59].

Requirements of Feedstock for Pyrolysis

One important economical constraint on the production of green diesel is the fact that the natural products used as feedstock have other important uses. A large number of studies on pyrolysis have been conducted on seeds that have higher consumption in the edible oil market [56,61], making them a less economical option. However, pyrolysis has been found to be highly flexible in its choice of feedstock, with researchers reporting promising results with respect to the use of waste materials in pyrolysis reactors [34]. The selection of appropriate feedstock remains critical, however, as there are several requirements with respect to the pyrolysis reaction [63].

For fast pyrolysis, the feedstock must have a water content less than 15% of the total mass of feedstock to ensure the energy efficiency of the pyrolysis oil [58]. Drying the feedstock by using natural heat sources such as the sun or the heat produced during other processes is important to reduce water content [63] Also, a lower ash content in the feedstock is important as it influences the yield of oil [64]. In catalytic fast pyrolysis, the ash must be even lower, as ash metals can result in the deactivation of catalysts through poisoning [63].

## 4. Type of Catalyst in DO

Several catalysts can be used to obtain green fuels from triacylglycerol feedstocks depending upon the method used. The use of a homogeneous catalyst is reported most commonly in studies using DO for the production of diesel through decarboxylation. This is despite the fact that research has found heterogeneous catalysts for decarboxylation of carboxylic acids to offer better performance than homogeneous catalysts [38]. The main benefit of heterogeneous catalysts is that they do not require the addition of extra mechanisms for their removal from diesel products, as they are usually solid and do not mix with the main liquid product. Furthermore, they reduce the required maintenance of reactors as these catalysts are not corrosive.

Kim et al. [55] conducted an analysis of the reaction pathway of DO with soybean oil and reported several important findings with respect to the choice of catalysts. They noted that the process of DO leads to production of partially deoxygenated intermediates such as alcohols and aldehydes,

and that, as carbon–oxygen bonding in alcohol is weaker than bonding in aldehydes, the CoMo and NiMo catalytic HDO can advance through hydrogenation of the latter to form alcohols and then hydrocarbons, releasing $H_2O$.

Kim et al. [55] used a $C_{17}/C_{18}$ alkane ratio to analyze the reactivities of various DO processes in the presence of different catalysts. They found that Ni- and Pd-based catalysts had higher alkane ratios than CoMo- and NiMo-based catalysts, showing that the DO process in the presence of former group of catalysts occurs through decarboxylation or decarbonylation, whereas the process in the presence of the latter group of catalysts is direct. Hence, when CoMo- and Ni-based catalysts are used, the reaction temperature does not influence the reaction pathway, but when Pd-based catalysts are used, an increase in reaction temperature leads to improved HDO rates. Similarly, increase in pressure was reported to have a positive effect on HDO rates for CoMo- and Ni-based catalysts [55].

The difference in DO reaction pathways between Mo-based and metallic catalysts lies with the reaction of carbon–carbon and carbon–oxygen bonds with those catalysts [55]. The process of DO breaks the carbon–carbon bond to produce $CO_2$ and CO and breaks the carbon–oxygen bond to produce $H_2O$. While HDO sequentially breaks the carbon–oxygen bond over metallic catalysts in two steps, the one-step breaking of carbon–carbon bonds by Mo-based catalysts produces relatively lower low C17/C18 ratios as compared to Ni- and Pd-based catalysts, indicating that breaking of carbon–carbon bonds involves decarboxylation or decarbonylation.

A prior study comparing the breaking of carbon–carbon and carbon–oxygen bonds in hydrocarbons also reported higher rates of carbon–carbon cleavage in ethanol than carbon–oxygen cleavage over Pt-based catalysts [65]. In another study, evidence was shown of an increased cleavage rate of carbon–oxygen bonds in the presence of the sulfhydryl group on the ends of transition metal sulfides [66]. Do et al. [67] conducted catalytic DO of methyl stearate and methyl octanoate over Pt-based catalysts and reported that $Pt/TiO_2$ was a better choice for the conversion of methyl octanoate to $C_8$ as compared to $Pt/Al_2O_3$, due to higher cleavage activity. They explained that the possible reasons for higher carbon–oxygen bond breaking activity in $Pt/TiO_2$ over $Pt/Al_2O_3$ included the higher reducibility of titanium in comparison to aluminum.

The use of metal catalysts for the DO of hydrocarbons has increased notably in recent decades, with multiple studies reporting successful approaches. The main reaction involves removal of the carboxyl group and its conversion to $CO_2$ or CO through carboxylation, thus yielding a linear-chained hydrocarbon after removal of a carbon from the actual fatty acid chain. The rationale for using metal catalysts is that the process does not involve hydrotreatment and, hence, it can be conducted without high pressure [68]. The metal catalysts tested in recent studies include Pd-catalysts [49–51], Ni-catalysts [38–41,68,69], and Pt-catalysts [67], and studies have reported on the relative importance of each of these for various methods of DO.

In the case of pyrolysis, zeolite catalysts are usually used [69], but when the hydroxygenation method is used, the most commonly used catalysts are NiMo or CoMo [43,46–48]. Veriansyah [70] compared the effects of different catalysts on the hydroxygenation of soybean oil and found NiMo to offer the highest conversion rate at a catalyst/oil weight ratio of 0.044 and the second highest conversion rate at a ratio of 0.088. Ni-based and Pd-based catalysts were also found to be highly effective in the production of green diesel through hydroxygenation. CoMo, in comparison, had only moderate conversation rates and resulted in much lower concentrations of straight-chain n-alkanes in the final product.

## 5. DO over Carbon-Based Catalysts

Important limitations of metal catalysts for the production of diesel through DO include high cost and limited availability. The search for more cost effective solutions to this problem has therefore led to the introduction of carbon-based catalysts for this process. Pioneering work on less costly metal catalysts was done by Jasinski [71], who established Co-phthalocyanines as a new catalyst group for the oxidation of hydrocarbons in 1964, resulting in a series of studies experimenting with other novel

metal catalysts [72–74]. Carbon-based catalysts were later introduced as a less costly option for DO of hydrocarbons, with the potential to effectively produce green diesel much more cheaply [75]. Research has also shown that carbon-based catalysts tend to be less energy-intensive in reactions involving metal reclamation, as other process based on pyrometallurgy often require very high temperatures (up to 1500 °C) for melting and separating the different metals when Ni- or Al-based catalysts are used [76].

Three important issues with respect to use of carbon-based catalysts thus require discussion. First, the synthesis of carbon-based catalysts poses a concern because there are several different methods proposed for the process based on different rationales with regard to how the catalysts break carbon–oxygen bonds. Second, the process of decarboxylation or decarbonylation selectivity over carbon-based catalysts requires attention. Third, the lower acidity of carbon decreases its carbon affinity and makes it a more suitable choice of catalysts in terms of lower maintenance cost. This carbon affinity is discussed at length late in this paper.

*5.1. Development of Carbon-Based Catalyst*

In the development of carbon-based catalyst, it has been agreed that chemical activation play main role in improving the physicochemical properties of catalyst and improve the reaction activity. Table 2 summarized the synthesis method use in developing effective carbon-base catalyst. The main reaction pathway in DO over carbon-based catalysts, as described in several studies, is that the metal–N4 moiety that is bonded with the carbon is the main actor in reducing oxygenation during the DO process [73,74]. It has been emphasized that carbon-based catalysts cannot be produced in the absence of nitrogen, with this moiety serving as the main site for the reaction. Rejecting this finding, however, Yeager [77] and Wiesener [78] reported that the transition metal is not an active reaction site for oxygen reduction, as suggested, but that its main function is to expedite the steady integration of nitrogen into the carbon structure during the pyrolysis of metal–nitrogen complexes. This is further explained by Maldonado and Stevenson [79], who state that the strong basicity of N-doped carbons helps in the adsorption of reductive oxygen gas as well as the decomposition of peroxides, allowing these carbons to increase the catalytic activity.

This idea has gained further strength from recent experimental studies wherein the process of oxygenation was successfully performed over CoN/C-based catalysts [80–82]. X-ray photoelectron spectroscopy (XPS) was conducted in these studies to examine the catalysis reaction, and it was observed that there was an increase in the concentrations of nitrogen functional groups on carbon when pyrolysis occurred in the presence of cobalt. However, the liquefying of cobalt metals from Co–N chelates was not found to produce any effect on catalytic activity, while catalytic activity increased when excess inactive cobalt species were removed, resulting in the revelation of the real active sites of reaction, particularly carbon [82].

The synthesis of nitrogen-containing carbon-based catalysts is usually conducted through implantation, a process involving $NH_3$ or HCN treatment of an oxidized carbon. However, one study found that this method is not very effective, as the onset potential of oxygen reduction on $NH_3$-treated Ketjen black is much higher than on untreated carbon [80]. When quantum mechanical calculations were used to study cluster models, it was revealed that the carbon radical sites present next to nitrogen in the nitrogen-treated carbon are the real active sites [83], which led to proposals to improve synthetic methods for producing carbon-based catalysts. In some studies [84–86], the function of the nanostructure in nitrogen-containing carbon-based catalysts for oxygen reduction was also studied. The result showed that high activity for oxygen reduction was accomplished; the reasoning behind this is that the higher activity of the catalysts is associated with higher quantities of pyridinic nitrogen [86].

Subramanian [87] also recently introduced an improved method for carbon-based catalyst synthesis for oxygen reduction. The process is composed of four steps. The first step is removal of metal impurities through prewashing, while the second step is chemical oxidation of carbon support by adding formaldehyde, melamine, or urea to the amino group. The third step involves the synthesis

of nitrogen-rich polymeric resins on the oxidized carbon by means of a condensation reaction, and the final step is pyrolysis at elevated temperatures of 400 to 1000 °C in an inert atmosphere. The DO reaction using this carbon-based catalysts was found to offer no metal impurities, proving that only nitrogen modified carbon-based structures are responsible for the observed catalytic activity. The study concluded that a high concentration of pyridinic type nitrogen groups doped on graphitic carbon increases the activity of catalysts and, hence, these are the main sites of reaction. The results were further confirmed by Oh et al. [88], who synthesized carbon-based catalysts using three different types of carbon, consequently discovering that the carbon structure is the main site for oxygen reduction reaction.

Other methods for synthesis of carbon-based catalysts have also been attempted. Shu et al. [89] introduced sulfonation of carbon-treated vegetable oil asphalts to synthesize carbon-based catalysts. These catalysts were found to have high catalytic activity with conversion rates as 94.8 wt. % with a catalyst/oil ratio of 0.2 wt. %. The catalysts were also very stable due to high acid site density and hydrophobicity. Toda et al. [90] synthesized carbon-based catalysts through the incineration of sugar; however, the catalytic activity in this case turned out to be very low at 50 wt. %. A major advantage of this method, however, was that the catalysts could be reused without any effect on catalytic activity. Using the sulfonation method, Dehkoda et al. [91] synthesized KOH/AC and used it as a carbon-based catalyst to produce biodiesel from palm oil; this was successful in yielding oil up to 94% wt. %. They also reported that the catalysts could be reused three times for such production. In the same research, the researchers produced renewable diesel from canola oil with a conversion rate of 92 wt. %. As can be seen, however, all these studies used feedstocks from the edible market, and there is thus a need to conduct further research on nonedible oils or oil waste products for economic reasons.

In a recent study, the synthesis of carbon-based catalysts from carbohydrates such as glucose and sucrose has been proposed [92]. This study was unique in terms of utilizing waste cooking oils to create biodiesel, and it reported that the carbohydrate-derived carbon-based catalyst exhibited higher catalytic activity than other carbon-based catalysts. The carbohydrate derived catalysts emerged as very stable and, under optimized reaction conditions, retained catalytic activity, producing 93% biodiesel even after 50 cycles of reuse. In another study, fructose was used to synthesize carbon-based catalysts; this was found to be more eco-friendly and to produce higher yield of 5-hydroxymethylfurfural (HMF), a feedstock for the production of green diesel [93]. The method was thus able to produce 91.2% yield of HMF as well as retaining catalytic activity for five times of reuse.

Alsultan et al. [94] synthesized CaO-La$_2$O$_3$/AC nanocatalysts from walnut shells by carbonizing the pretreated shell powder at 400 °C for 4 h. Phosphoric acid was used to activate the carbon, and the pH was maintained at 7 through distillation. This active carbon was used to synthesize carbon-based metal catalysts by impregnation with La and Ca salt, and after 6 hours soaking, was dried overnight at 100 °C. The metal and active carbon solution was then heated to 700 °C for 4 hours. This modified reaction pathway was designed to balance the acidic and basic properties of the catalysts.

**Table 2.** Summary of carbon-based catalysts development studies.

| Method | Catalyst | Carbon Source | Synthesis Method | Surface Area ($m^2g^{-1}$) | Remarks | Ref. |
|---|---|---|---|---|---|---|
| N-containing carbon | $FeN_x/C$ $CoN_x/C$ $CoFe_3N_x/C$ $Co_3FeN_x/C$ $CoFeN_x/C$ | Carbon black | Co–N or Co–Fe–N chelate complex on the carbon support, | - | N-treated carbon are the real active sites | [79] [80] [81] |
| | Vulcan carbon (VC) Fe/VC Ni/VC | Vulcan carbon | Vulcan carbon impregnated with 2 wt % Fe or 2 wt % Ni | 228–234 | N-containing carbon samples in nanostructures resulting in exposure of more edge planes and lead to high activity | [86] |
| | C UF-C SeUF-C | Carbon black | Nitrogen-rich polymeric resins on the oxidized carbon. Resins: melamine formaldehyde (MF), urea formaldehyde (UF), thiourea formaldehyde (TUF), and selenourea formaldehyde (SeUF) by a simple addition-condensation reaction on the oxidized carbon | 321–694 | On the carbon surface, pyridinic (quaternary) and graphitic nitrogens act as catalytic sites for oxygen reduction | [87] |
| Sulfonation | V–C-600–S-210 | Vegetable oil asphalt | Carbonized vegetable oil asphalt (5.0 g) and 100 mL concentrated $H_2SO_4$ (96%) | - | high catalytic activity and stability related to its high acid site density (–OH, Brönsted acid sites) | [89] |
| | Sulfonated C | D-glucose sucrose | Sulfonate incompletely carbonized saccharides | - | High stability of catalyst | [90] [92] |
| | Cat A1 Cat A2 Cat A3 | Biochar | Sulfonation by $H_2SO_4$ followed by KOH chemical treatment | 2–6 | High stability of catalyst | [91] |
| Phosporus-metal dual doped carbon | $CaO-La_2O_3/AC_{nano}$ | Walnut shell | AC treated by phosphoric acid followed by incorporation of Ca and La species via vacuum impregnation method | 150–223 | Balance acid–base sites formed | [94] |

*5.2. Decarboxylation/Decarbonylation Selectivity over Carbon-Based Catalyst*

The reaction pathways of decarboxylation and decarbonylation are critical to affect the quality of green diesel due to their influences on the acidity and heating value of the diesel, respectively. As decarboxylation and decarbonylation are overlapping processes in the production of green diesel, the catalysts used should be the same. Existing research has mainly presented the use of $Al_2O_3$, $SiO_2$, $TiO_2$, or MgO-supported transition/noble metals such as Pt, Pd, Cu, or Ru as the most useful catalysts for decarboxylation [59], and few studies only reported on decarboxylation or decarbonylation over carbon-based catalysts. Table 3 showed several studies in DO of vegetable oil or fatty acid derivatives over carbon-based catalysts. Noble metal (Pd) rendered the most effective active promoter that incorporated on carbon support for enhancing the DO activity via decarboxylation/decarbonylation pathways [95–97]. Interestingly, metal promoter (Mg) with basicity properties supported on carbon catalyst also effective in removing the oxygenated bonded compound in palm oil conversion [11]. However, the MgO/C showed least DO activity (65% hydrocarbon yield and 12% n-$C_{15}$ selectivity) when compared with other studies in Table 3. Although palm oil composed of 45% palmitic (C16) and 55% oleic (C18), but only n-$C_{15}$ is formed and no peak was detected at n-$C_{17}$, which suggested that certain reactions of carbon–carbon dissociation occurred.

Alsultan et al. [94] identified that the use of basic metallic catalysts has been suggested in recent years mainly because of their higher stability; however, these catalysts often require hydrogen supplies for the DO process, as they tend have lower decarboxylation and decarbonylation selectivity. In order to resolve this problem, he suggested the use of carbon-based catalysts that can provide both acidic and basic metallic properties. He tested CaO-$La_2O_3$/AC nanocatalysts for the DO of waste cooking oil under an interreaction atmosphere and obtained yields of 72% hydrocarbons through decarboxylation and decarbonylation. The catalysts were also very stable and retained their conversion rate and hydrocarbon selectivity for up to six cycles. Recently, Alsultan and coworker also investigated green diesel production via DO reaction of waste cooking oil over $Ag_2O_3$–$La_2O_3$/$AC_{nano}$ catalyst under a hydrogen-free environment [98]. It was revealed that the $Ag_2O_{3(10\%)}$–$La_2O_{3(20\%)}$/$AC_{nano}$ formulation resulted in a higher yield (~89%) of liquid hydrocarbons with majority of diesel fractions selectivity (n–($C_{15}$+$C_{17}$) at ~93%. The high stability of the $Ag_2O_{3(10\%)}$–$La_2O_{3(20\%)}$/$AC_{nano}$ catalyst was proven by maintain six continuous runs with constant yield (>80%) of hydrocarbons and (>93%) selectivity of n–($C_{15}$+$C_{17}$) under mild reaction conditions. It can be suggested that the use of a carbon support in a nanosized structure offers great tendency in promoting the DO reaction with reaction selectively toward decarboxylation and decarbonylation pathways (figure 4).

Asikin-Mijan [99] used multiwalled carbon nanotube (MWCNT)-based catalysts for the catalytic DO of *Jatropha curcas* oil and reported similar high selectivity decarboxylation and decarbonylation. The oil was deoxygenized in a semi batch reactor with 5 wt % of catalysts. Analysis of the deoxygenized oil showed that the catalyst activity mainly occurred through decarboxylation and decarbonylation paths, and the hydrocarbon yield reached over 80% with the use of Ni/MWCNT catalysts. However, Ni-Co/MWCNT catalysts showed the highest selectivity to $C_{15}$ and $C_{17}$ hydrocarbons, suggesting that the presence of acidity results in selective mild cracking of the triglyceride structure and, therefore, that the catalysts perform actively in terms of DO by means of decarboxylation and decarbonylation. Overall, from the DO reaction over carbon supported catalyst summarized that acid and acid–base promoted catalyst effectively promote the removal of oxygenates compound via decarboxylation and decarbonylation reaction than base promoted catalyst [100].

**Table 3.** Summary of the carbon-based catalysts in DO via the decarboxylation/decarbonylation reaction.

| Catalyst | Reaction | Feed | Reaction Condition | Reactor Mode | H/C Yield (%) | Diesel Selectivity (%) | Coke (wt %) | Ref. |
|---|---|---|---|---|---|---|---|---|
| Pd/C, Pd/Sebunit | DO | Lauric acid | 26 h, 300 °C: 0.075 ml/min reactant flow rate (WHSV 0.33h$^{-1}$), 15 bar Ag, 10 ml/min argon flow, 4.4 mol/l (solvent-free conditions). | Fixed-bed | >80 | - | - | [38] |
| Pd/C | DO | Lauric acid | 1 h; 270 °C; Atm pressure | Fixed-bed | - | - | - | [96] |
| Pd/C, Pt/C | DO | Stearic acid | 6 h; 300 °C, 6 bar; He 25ml/min | Semibatch | >90 | 98 | - | [101] |
| MgO-C | DO | Palm oil | 430 °C, feed late 13.5 g/h., under atm pressure | Semibatch | 65 | 12 | - | [11] |
| Ni/MWCNT NiCo/MWCNT | DO | JCO | 1 h; 350 °C; partial vacuum condition | Semibatch | 82 | 48 | - 4.5 | [99] |
| CaO-La$_2$O$_3$/AC$_{nano}$ | DO | WCO | 3 h; 330 °C; N$_2$ flow condition | Semibatch | 73 | 82 | 1.5 | [101] |
| Ag$_2$O$_3$–La$_2$O$_3$/AC$_{nano}$ | DO | WCO | 2 h; 350 °C; N$_2$ flow condition | Semibatch | 89 | 93 | 1.8 | [98] |

### 5.3. Coke Affinity over Carbon-Based Catalyst

The coke affinity of metallic catalysts has been one major reason behind the search for more stable catalysts for DO [69,100,102]. A coked catalyst is detrimental to the process, as it may lead to severe plugging of the reactor or even to catalytic particle disintegration if the coking effect is left untreated. The type and nature of any carbonaceous deposits are dependent on the type of catalyst, reactant, and reaction. With regard to the catalyst, an increase in acidity tends to promote the formation of carbonaceous deposits.

Coke affinity over carbon-based catalysts during DO of natural feeds has been studied previously, when they were found to have high tolerance for coke colonization [94,98,99] (Table 3). Alsultan et al. [89] reported that the coke formation in DO of waste cooking oil over carbon-based catalysts at pH 7 was less than 1.5 wt % and 1.8 wt % (Figure 4D) after six times of reuse, which is trivial and can thus be ignored. The degree of coke formation was ascertained through thermogravimetric analysis (TGA). Asikin-Mijan et al. [101] similarly used Ni-Co/MWCNT catalysts to deoxygenate vegetable oil for the production of biofuel. Under an inert atmosphere with no hydrogen, the catalysts were reused for DO more than four times, and the amount of coke formed during the process was negligible. It is clear that carbon-based catalysts have much higher resistance to coke formation and, hence, are more stable than other metallic catalysts [103,104].

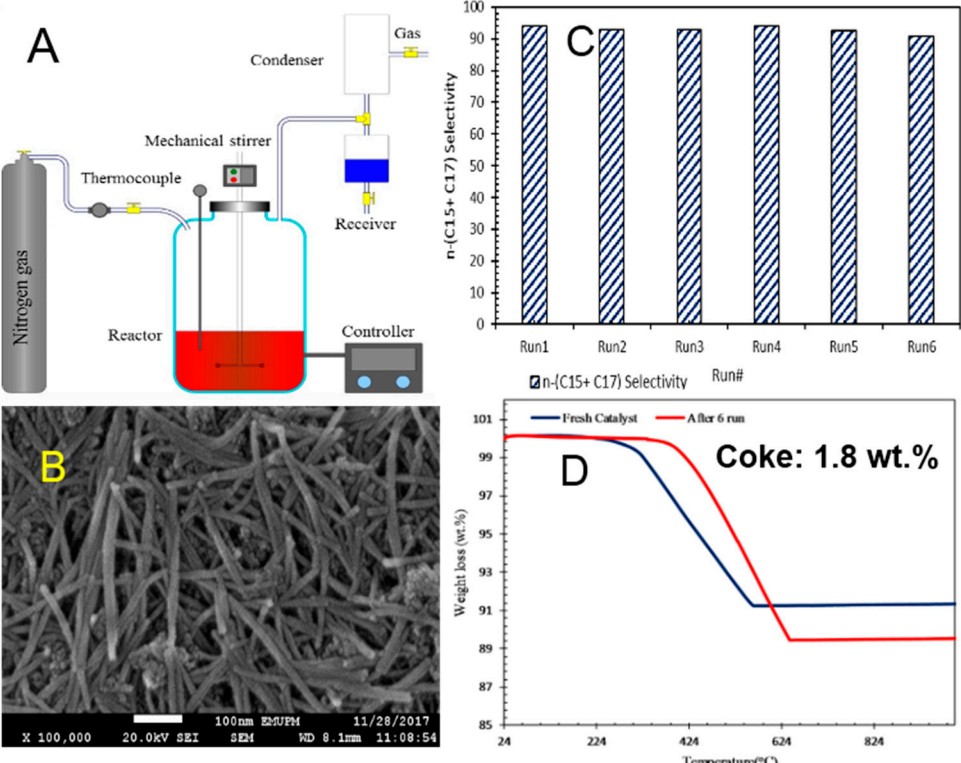

**Figure 4.** (**A**) Schematic diagram for the DO set-up, (**B**) FESEM image for $Ag_2O_3$–$La_2O_3/AC_{nano}$, (**C**) n-($C_{15}$+$C_{17}$) selectivity with the number of DO cycles at temperature of 350 °C, catalyst amount of 1 wt % and reaction time of 120 min under inert condition, and (**D**) thermogravimetric analysis (TGA) profile for fresh and spent $Ag_2O_{(10\%)}$-$La_2O_{3(20\%)}/AC_{nano}$ catalyst after six runs [98].

### 6. Factor Influencing the DO Process

The main parameter for measuring the effectiveness of the DO process is the concentration of diesel-like hydrocarbons in the final product, which reduces the cost and effort of separation of the diesel from contaminants. A number of factors influence this concentration, but a comprehensive inquiry into all these factors has only been done by few scholars, and more work is needed in this area [7].

Three important factors have been discussed, however, which have been proved through multiple studies to influence the diesel content: feedstock, reaction atmosphere, and reaction temperature.

*6.1. Feedstocks*

The choice of feedstock can significantly influence the DO process. For the production of green diesel, the most commonly used feedstocks are plant derived oils such as soybean, rapeseed, and palm [23,36,55,56]. However, as discussed above, these oils have strong edible markets and alternatives, such as nonedible oils (jatropha and algal oils) and waste oil products, thus become more popular in recent years [56,61]. There is also some debate over the use of saturated fatty acids and unsaturated fatty acids as feedstocks.

When nonedible oil sources are used, the hydrocarbons are synthesized through a process called Ecofining [18]. This process is robust to high concentrations of free fatty acid, allowing the less costly nonedible oil sources to be used as feedstocks. The concentration of fatty acid in feedstocks is a defining feature, as the high concentrations of saturated fats in palm and tallow oils require significantly less hydrogen. In comparison, when soybean and rapeseed oils are used to feed the DO process, their higher olefin content makes it necessary for additional hydrogen to be added to the process. With respect to the concentration of certain waste products in all cases, it is usually required to prewash these feedstocks for removal of solids and salt [18,105].

Snåre et al. [38] also compared the differences in yields for various natural sources feeding the DO process. They reported that diunsaturated linoleic acid exhibited a lower conversion rate in comparison to monounsaturated oleic acid and methyl oleate, explaining that presence of unsaturated fatty acids required an additional hydrogenation step to convert these to monounsaturated fatty acids before they could be converted to the desired hydrocarbons. In addition, the presence of unsaturated hydrocarbons can cause adsorption on the catalyst surface, leading to catalyst deactivation and higher likelihoods of cracking and coking reactions [35].

Other studies have also reported that feedstocks with lower concentrations of unsaturated fatty acid require less hydrogen input [18,41]. This is because unsaturated fatty acids have been reported to reduce selectivity to *n*-alkane [106]. Hence, research suggests that DO of fatty acid results in n-alkane selectivity of 86%, while DO of fatty acid esters yields a selectivity of only 40%. This has been attributed to higher rates of decarboxylation in fatty acid DO compared to fatty acid ester DO [106]. The higher robustness of the DO process leads to a lower diesel cloud point with the drawback of a yield shift from diesel to lower molecular weight fuels. However, the Ecofining process allows changes in operating conditions to allow the choice of feedstock as well as sporadic alternations in diesel cloud point specification [18].

*6.2. Reaction Atmosphere*

Several studies have compared differences in yield with respect to reaction atmosphere, particularly with respect to presence of hydrogen [38,69,103–108]. Rozmysłowicz et al. [107] examined the effect of hydrogen in the reaction atmosphere in terms of the yield of desired hydrocarbons in lauric acid DO. Using a semibatch reactor, they compared the yield of hydrocarbons through DO over Pd/C catalysts under inert atmosphere with that under a hydrogen-rich atmosphere. The experiment showed that, for the initial 100 min, the inert atmosphere was more favorable, yielding a much higher proportion of desired hydrocarbons in comparison to the hydrogen rich environment. This was explained as being caused by the formation of intermediates in the hydrogen rich atmosphere, which were later converted into the desired hydrocarbons. Hence, after this initial time, the conversion rate of DO conducted in the presence of hydrogen became significantly higher. The main difference is the reaction pathway of the two processes: hydrogen presence leads to hydroxylation and its absence in the inert atmosphere supports decarboxylation and decarbonylation.

Later studies examining the effects of hydrogen in the reaction atmosphere have revealed that, in comparison to both inert atmosphere and hydrogen rich atmosphere, a small amount of hydrogen

in an otherwise inert atmosphere yields a much higher concentration of hydrocarbons [38,69,100]. Snåre et al. [38] compared the DO of oleic acid over Pd/C under various conditions and reported that the presence of hydrogen resulted in an increase in conversion rate when compared with an inert atmosphere. Lestari et al. [100] tested the effect of hydrogen in the reaction atmosphere on DO of stearic acid and palmitic acid, while Madsen et al. [69] conducted research on dilute and concentrated steric acid. The higher yield was explained as being caused by the smaller amount of $H_2$ in the reaction atmosphere, which generated lower amounts of unwanted unsaturated hydrocarbons and produced the required saturated hydrocarbons. In addition, the presence of a small amount of hydrogen results in reduced coke formation, preventing catalyst deactivation in as seen in the inert environment. Thus, the addition of a small quantity of $H_2$ to the inert gas was found to maintain the catalytic activity of Pd/C catalysts by avoiding coke formation [69,100].

Pattanaik and Misra [35] concluded, after a review of studies examining the effect of reaction atmosphere on DO, DO pathways differ with respect to reaction atmosphere. Under an $H_2$-rich atmosphere, the reaction pathways are similar irrespective of the choice of feedstock; however, under an inert gas atmosphere, reaction pathways and product selectivity differ significantly between fatty acids and esters and triglyceride feeds, with fatty acids and esters exhibiting much lower conversion rates and catalyst stability due to cracking. Hence, for these feeds, an $H_2$+Ar environment is preferred.

### 6.3. Reaction Temperature

The usual reaction temperature range for DO is 250 to 360 °C, and studies that have reported notable changes in the yields of diesel-like hydrocarbons with changes in reaction atmosphere are generally conducted within this range [4,35,38,94,105]. Snare et al. [38] examined the effect of reaction temperature on oleic acid DO from 300°C to 360 °C, and the conversion rate was reported to be 78% at 300 °C, increasing to 93% at 330 °C and 360 °C. Bernas et al. [4] studied the effect in diluted dodecanoic acid DO in the same temperature range and reported an increase in conversion to undecane and undene from 10% at 300 °C to 60% at 360 °C. In another study, the conversion rates of diluted stearic acid at 270 °C and at 330 °C were compared, and it was reported that there was a significant decrease in time for 100% conversion to the desired hydrocarbons with this increase [104]. The conversion of diluted ethyl stearate to diesel fuel through DO in a semibatch reaction also increased from 40% at 300 °C to 100% at 360 °C [50]. The researchers noted that the main reason for this increase was the reduction in selectivity of n-heptadecane from 70% to 40% in the same temperature range, indicating that high temperature results in a decrease in selectivity of diesel-like hydrocarbons.

A change in yield of diesel with respect to change in reaction temperature of DO tall oil fatty acid (TOFA) over Pd/C catalyst from 300 °C to 350 °C has also been reported [40]. The experiment results showed that selectivity of diesel fuel decreased with increase in reaction temperature, and only a slight coke formation of Pd/C catalysts was observed at the highest reaction temperature of 350 °C. This shift occurs mainly in the selectivity of the hydrocarbons, with higher yields of n-heptadecane compared to n-heptadecene at 300 °C and relatively lower ones at higher reaction temperatures of 350 °C [105].

Na et al. [89] examined the effect of reaction temperature on the decarboxylation of oleic acid in an $H_2$-free environment at a temperature range of 350 °C to 400 °C. They reported more than 98% conversion at 400 °C over three different hydrotalcite catalysts, and noted that at lower temperatures, saponification of oleic acid and MgO (catalyst) resulted in the formation of solid-like products that affected the conversion rate. When oleic was deoxygenized under a $H_2$-rich atmosphere, the rise in temperature similarly resulted in an increase in conversion rate [69]. This shows that the effect of reaction temperature occurs irrespective of catalyst choice and reaction atmosphere and, hence, is not dependent on the reaction pathway for DO.

Overall, a rise in reaction temperature tends to result in an increase in the conversion rate of DO while decreasing DO selectivity. Pattanaik and Misra [35] explained that this is mainly due to the increased rate of thermal decomposition at higher temperatures and the synthesis of additional aromatic products. The solution to this is to define a balanced approach with an optimized temperature

range that produces good conversion rates with adequate DO selectivity. They also explained that reaction temperature does not influence the reaction pathway of DO, which is mainly dependent on the choice of catalyst.

## 7. Recent Work on the Deoxygenation of Realistic Oil

The DO of realistic oils (safflower, soybean, jatropha, and karanja) has established itself as a potential resource for the production of green and biofuels for petroleum fuel replacement. In the past, the research focus has been establishing the significance of DO of vegetable oils as a commercial technology for hydrocarbon fuel production. One major concern is the high cost of green diesel produced through DO of realistic oils. In recent years, the studies on portentous ways to make green fuel more feasible and commercially friendly. Thus, studies have been conducted on the use of low-cost catalysts, waste products as feedstocks, and $H_2$-free reaction atmospheres. Another important factor in recent research has been focused on the improvement of fuel quality, not just in terms of increasing the yield and selectivity of the hydrocarbon fuel, but also in terms of the reusability of the catalysts. Some of the state of the art research was briefly introduced to highlight current trends in the DO technology, which serve as guidance for further research study.

The cost effectiveness of biofuels was the major concern for Meller et al. [106], who with the DO of castor oil FAME over Pd/C catalyst with high yield of hydrocarbons mainly comprised of n-heptadecane. In order to reduce the cost of production, they conducted their research under $H_2$-free environment, which reported that the main reaction pathway for the DO of castor oil FAME was decarboxylation and decarbonylation, with methyl stearate and stearic acid serving as intermediators. Yeh et al. [107] also used a $H_2$-free environment of DO process for lower process cost when comparing the reactivity of three different Sn containing catalysts ($Pt_3Sn/C$, $PtSn/C$, and $PtSn_3/C$) towards yield and hydrocarbon selectivity. The finding indicated that $PtSn_x/C$ as having a much higher yield than the other catalysts under similar conditions.

Sergiy Popov and Sandeep Kumar [108] investigated converting oleic acid into n-alkane fuel under a continuous flow process using supercritical water as the reaction medium, granulated activated carbon as a catalyst, and 1% v/v formic acid as an in situ source of hydrogen. Experiments were conducted in a packed tubular reactor with the weight hourly space velocity of $4 \ h^{-1}$ at temperatures from 350 to 400 °C and pressure 3500 psi (24.1 MPa). The oil to water to formic acid ratio was 1:5:0.05 by volume. The main reaction pathways were hydrogenation of oleic acid and decarboxylation/decarbonylation of the resulting stearic acid to form heptadecane. The yield of heptadecane was above 70% with a selectivity of 80%, observed between 370 and 380 °C. The results of the study show that efficient hydrothermal deoxygenation of fatty acids can be achieved with activated carbon as a catalyst and formic acid as an in-situ source of hydrogen within minutes. Kinetics study showed that the rates of oleic acid conversion displayed Arrhenius behavior with an activation energy of 120 kJ/mol.

Recent studies also focused on the use of alternative feedstocks due to vegetable oils having high rates of consumption in the edible market. Yeh et al. [107] compared the use of saturated and unsaturated feedstock and found the former to be a better choice for decarboxylation/decarbonylation to green diesel product. Silva et al. [109] introduced macauba almond oil as a nonedible feedstock, reporting that catalytic DO of hydrolyzed macauba almond oils over Pd/C catalyst produced linear saturated hydrocarbons, which suitable for to use as green diesel. They also found that the main reaction pathway was decarboxylation/decarbonylation and that an intensive hydrogen supply was therefore not needed for the reaction. Using a 300 °C reaction temperature, 5 h reaction time, and 700 rpm stirring speed, they were able to achieve yields of up to 85 wt % of diesel-like hydrocarbons.

Le Kim Hoang Pham et al. [16] investigated the hydrodeoxygenation (HDO) of waste cooking oil (WCO) in a continuous fixed-bed reactor over a series of activated carbon (AC)-supported nickel phosphide catalysts with different initial Ni/P molar ratios (0.5–2.0) and nickel loading levels (1.16–38.90 mmol/g AC). The formation of the Ni-P phase on the AC was produced from commercial charcoal as well as its structural and acidic properties. The effects of the Ni/P molar ratio, nickel

loading level, reaction temperature, and gas hourly space velocity (GHSV) on the catalytic activity were elucidated. The complete formation of the Ni-P phase on the AC was observed at a Ni/P ratio of 1.5, while smaller Ni-P crystallite sizes were observed at lower Ni/P ratios. In addition, they observed that the acidity increased and the specific surface area decreased with an increase in the nickel loading level, presumably because nickel phosphate is not readily reduced to Ni 2 P/1.5TPR catalyst (Ni loading level of 5.37 mmol/g AC and Ni/P molar ratio of 1.5) exhibited good activity and stability during the HDO of WCO. The high-quality deoxygenated product primarily consisted of n-alkanes at the moderately high temperature of 300 °C and GHSV of 2.33 min$^{-1}$. Le Kim Hoang Pham's study [16] proposed mechanism underlying the hydrotreatment of WCO involves hydrogenolysis, hydrodeoxygenation, dehydration decarbonylation, and hydrogenation.

Simon Eibner et al. [110] demonstrated the potential of activated charcoal-based catalysts with weak acidic properties that partially deoxygenated bio-oils feedstock. The group successful to develop an innovative method to synthesize activated charcoal-based catalysts doped with $CeO_2$, $Fe_2O_3$ or $Mn_3O$ nanoparticles. They investigated the performances of those catalysts to deoxygenate two biomass pyrolytic model compounds: acetic acid and guaiacol on a fixed-bed reactor between 350 °C and 450 °C. A ceria-based catalyst was highly active and remarkably stable to enhance ketonic decarboxylation of acetic acid, leading to the formation of acetone. Huge amounts of produced phenol attest for the partial deoxygenation of guaiacol, particularly when using iron-based catalyst.

The use of algae as DO feed to produce green diesel was introduced by Viegas et al. [111]. They conducted experiments on the DO of Chlorella algae over Ni-HY-80 zeolite and Pd/C catalysts using dodecane as a solvent at 60 and 100 °C. They reported higher catalytic activity in the Ni-HY-80 catalyst in comparison to the Pd/C catalyst, resulting in a greater than 95% yield of hydrocarbons. However, the green diesel produced through this process was found to lack of oxidative stability, and an additional hydrogenation process was suggested to improve stability of the biofuel. This is therefore not a feasible process, as other studies have provided better alternative feeds with much higher stability of diesel. Knoshaug et al. [98] also examined the use of algal sugars in the production of green diesel, using biomass from the green alga *Scenedesmus acutus* to produce succinic acid; the lipids produced in this reaction as a by-product, with 83% yield, were deoxygenized through HDO and hydroisomerization to produce green diesel.

In the search for improvements of DO process in terms of increased hydrocarbon yield and selectivity, as well as catalyst stability, several studies have been conducted in recent years. Wang et al. [112] compared the use of continuous and fed-batch reactors for liquid-phase DO of fatty acids derived from canola oil over Pd/C catalyst. Utilizing HDO as the main reaction pathway, the experiment yielded 99% n-alkane in fed-batch and 88% in continuous DO processes. In a continuous DO reactor, higher $H_2$ partial pressures were also found to improve the conversion rate. The researchers suggested that once hydrogenation is complete, less than 5% of effective $H_2$ partial pressure should be maintained to support the decarboxylation/decarbonylation pathway and to reduce the cost of reaction.

Aslam et al. [113] introduced new catalysts for hydroprocessing of high FFA *Mesua ferrea L.* and *Jatropha curcas* oil, namely the Musa balbisiana Colla underground stem (MBCUS), a nanomaterial, and a biomass-based thermal power plant fly ash (BBTPFS). Reaction pathway analysis showed that the production of the desired hydrocarbons in $C_{18}$–$C_{19}$ occurred through decarboxylation, hydrolysis, hydrocracking, and thermocatalytic cracking. Comparisons of conversion rates, catalyst acidity, and heteroatom removal also showed BBTPFS to be a better choice than MBCUS for the described process.

## 8. Conclusions

The search for more choices of sustainable fuel has been motivated by scientist striving to develop more feasible and environmentally protective solutions towards the problem of growing global demand for fuel. Green diesel has attracted much attention due to its potential fuel properties, environmental benefits, biodegradability, energy stability, and production cost. With regards on fuel properties study, it was noted that green diesel has an advantage over petrodiesel in terms of protecting environment

and biodegradability. In comparison to biodiesel, green diesel offers much higher stability, heating value, and exhaust emission of certain pollutants.

Although green diesel production cost under a debate but typically the green diesel production cost is highly dependent on the synthesis process used. It has been noted that removal of oxygenates bonded compound via DO pathway under hydrogen-free atmosphere are economically viable than HDO and pyrolysis. HDO is costly due to the high consumption of hydrogen gas during the process. The pyrolysis process has lower cost, but the hydrocarbon product mainly composed of light fractions.

Ni, Pd, Pt, Mo, NiMo, and CoMo are the common active promoters that apply as catalyst for DO of triacylglycerol to diesel range hydrocarbons. The effectiveness of the metal promoter can be improved by an additional functional group incorporated with metal promoters and the promoter should pose higher reducibility properties. The rationale for using metal-based catalysts is that the process does not involve hydrotreatment and, hence, it can be conducted under hydrogen-free atmosphere.

Metal-supported on carbon catalysts have been introduced in recent research since it could offer better DO activity and product selectively toward decarboxylation and decarbonylation pathways for producing high yield of diesel range hydrocarbon. The use of carbon as a catalyst support beneficial in reducing the green diesel production cost due to high availability of carbon sources. Carbon-based catalysts show better catalyst stability and high DO reactivity. This is due to its mild acidity properties simultaneously reduce its carbon affinity (reduce it tendency from coke formation).

Several studies were carried out to improving the physicochemical properties of carbon-based catalyst. In most cases, acid or acid–base properties are extremely important in enhancing the removal of oxygenates via carbon–oxygen dissociation under decarboxylation and decarbonylation reactions. Basicity sites alone are not preferable in the DO reaction, since it rendered high occurrence of carbon–carbon dissociation and reduce diesel hydrocarbons fractions.

Three important operating factors influencing production of diesel-range hydrocarbon via DO process are feedstock, reaction atmosphere, and reaction temperature. It was discovered that DO process under $H_2$+Ar environment preferable for fatty acids and esters as feedstocks. The use of triglycerides as a feedstock has also found much support from the literature, as they show higher conversion rates in hydrogen-free environments. Lower content of unsaturated fatty acid feedstock will be effective for DO reaction as compared to high content of unsaturated fatty acid feedstock such as ester. Regarding reaction temperature, a balance must be maintained with respect to selectivity of fuel-range hydrocarbons and the conversion rate of the feedstock.

Based on the recent works in DO of realistic oil, it can conclude that the utilization of carbon-based catalyst render high effectiveness in producing the diesel range hydrocarbons. Additionally, the use of nonedible oils, such as waste cooking oil, not only reduces the production cost, but can also potentially replace edible oil-based green diesels soon. Besides, the green diesel derived from waste cooking oil consisted of low volatile compounds compared to petroleum—the USLD standard. Thus, the waste cooking-based green diesel is much safer to be handled and apply in the diesel engine.

**Funding:** This research received no external funding.

**Acknowledgments:** The authors acknowledge the financial support from the PUTRA grant-UPM (Vot No: 9344200), MOSTI-e Science (Vot No: 5450746), Geran Putra Berimpak (GPB)UPM/800-3/3/1/GPB/2018/9658700, and the University of Malaya's RU grant (Project No:RU007C-2017D and ST012-2018).

**Conflicts of Interest:** The authors certify that they have no affiliations with or involvement in any organization or entity with any financial interest (such as honoraria; educational grants; participation in speakers' bureaus; membership, employment, consultancies, stock ownership, or other equity interest; and expert testimony or patent-licensing arrangements), or non-financial interest (such as personal or professional relationships, affiliations, knowledge or beliefs) in the subject matter or materials discussed in this manuscript.

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
