# Peer review of "A Review on Thermal Conversion of Plant Oil (Edible and Inedible) into Green Fuel Using Carbon-Based Nanocatalyst"

_catalysts, doi:10.3390/catal9040350_

Round 1

Reviewer 1 Report

In the manuscript entitled "Thermal Conversion of Plant Oil (Edible and Inedible) Into Green Fuel Using Carbon-Based Nanocatalyst", the authors reviewed recent research of carbon-based catalysts for fuel production from plant oil conversion. This is an interesting review and the topic of this manuscript fits Catalysts well. However, this manuscript has some drawbacks, and in my opinion the manuscript needs a major revision before publication. Followings are my concerns.

1. There are many formatting errors. For example, “H2-free” in line 21, “NiMo/g-Al2O3 or CoMo/gAl2O3” in line 173, “NH3” in line 325, “used in DO, HDO, and Pyrolysis” in line 675, etc.

2. All references in main text are not cited in correct order. For example, “as shown in table 1 [78].” in line 58, “this problem is the increased use of waste oil for the production of green diesels [42].” in line 42, “in figure 3 [69]” in line 203, etc.

3. The resolution of figure 2 should be improved. The reference of figure 3 should be given in the caption, if it is from published article.

4. It will be helpful to use tables to summarize research reviewed in section 5.

5. Section “7. Recent work on the deoxygenation of realistic oil” should also focus on carbon based catalysts.

6. Section “8. Conclusion” can be improved to be more related to the title of manuscript.

7. References format should follow the standard reference style of the journal.

Author Response

Thank you for your comment, all the answers can be found on the attached file

Reviewer 2 Report

This review article provides a comprehensive summary about various aspects of green diesel production such as reaction pathways and conditions, catalysts and feedstocks. An overview of the progress so far, including significant results are described.  

1. Introduction part may be elaborated.

2. The manuscript will be more complete if the authors can provide one or two         more figures. For example, a figure showing different types of carbon used. a      figure showing some conversion/selectivity profiles, etc.

3. Reference format need to be corrected. For example, in (3) Feb 15 is given        along with year. Similar corrections in (22), (40), (41), (47), (52), (70), (71),        (78), (88), (89), (108).   

Author Response

(The authors gave the same response as above.)

Reviewer 3 Report

The work presents a review, after consulting a  large number (108) of articles (53% of them in the last decade). It is an interesting article, and it shows an exhaustive search of fundamental information on the selected area. However, considering that this is a review article, the title must be changed, including "review" in it.  This will prepare readers, from the very beginning, to enjoy the article. 

Moreover, the abstract and conclusions must be modified and the primary authors' findings (after checks all reported references) must be included, general comments must be avoided, as well. 

Best Regards,

Author Response

(The authors gave the same response as above.)

Round 2

Reviewer 1 Report

Thanks for the modification and clarification from the authors. The present version has addressed most of the reviewer's concerns. I would like to recommend the manuscript for publication.